# *Starch Synthesis-Related Genes* (*SSRG*) Evolution in the Genus *Oryza*

**DOI:** 10.3390/plants10061057

**Published:** 2021-05-25

**Authors:** Karine E. Janner de Freitas, Railson Schreinert dos Santos, Carlos Busanello, Filipe de Carvalho Victoria, Jennifer Luz Lopes, Rod A. Wing, Antonio Costa de Oliveira

**Affiliations:** 1Centro de desenvolvimento Tecnológico—CDTec, Graduate Program in biotechnology, Capão do Leão Campus, Federal de Pelotas, Pelotas 96160, Brazil; karinejanner@gmail.com (K.E.J.d.F.); carlosbuzza@gmail.com (C.B.); jenniferlopesagronomia@gmail.com (J.L.L.); 2Instituto Federal de Educação, Ciência e Tecnologia Farroupilha (IFFar), Alegrete 97541, Brazil; railson.schreinert@iffarroupilha.edu.br; 3Núcleo de Estudos da Vegetação Antártica—NEVA, Campus São Gabriel Federal do Pampa (UNIPAMPA), São Gabriel 97030, Brazil; filipevictoria@unipampa.edu.br; 4The School of Plant Sciences, Ecology & Evolutionary Biology, Arizona Genomics Institute, Tucson, AZ 97030, USA; rwing@ag.arizona.edu; 5Center for Desert Agriculture, King Abdullah University of Science & Technology, Thuwal 23955, Saudi Arabia

**Keywords:** *Leersia perrieri*, phylogeny, starch synthesis, cooking quality

## Abstract

Cooking quality is an important attribute in Common/Asian rice (*Oryza*
*sativa* L.) varieties, being highly dependent on grain starch composition. This composition is known to be highly dependent on a cultivar’s genetics, but the way in which their genes express different phenotypes is not well understood. Further analysis of variation of grain quality genes using new information obtained from the wild relatives of rice should provide important insights into the evolution and potential use of these genetic resources. All analyses were conducted using bioinformatics approaches. The analysis of the protein sequences of grain quality genes across the *Oryza* suggest that the deletion/mutation of amino acids in active sites result in variations that can negatively affect specific steps of starch biosynthesis in the endosperm. On the other hand, the complete deletion of some genes in the wild species may not affect the amylose content. Here we present new insights for *Starch Synthesis-Related Genes (SSRGs)* evolution from starch-specific rice phenotypes.

## 1. Introduction

Common rice (*Oryza sativa* L.) is a food of great importance worldwide, especially in Asian countries, where it is an important part of local culture. Being widely consumed and having different forms of preparation makes “quality” something different in each country around the world. Nevertheless, no matter what grain quality means, its demand is increasingly becoming a priority for international export markets worldwide.

Today, cooking has become one of the most important research components in several rice breeding programs, where characteristics such as amylose content (AC) and gelatinization temperature (GT), which have major effects on cooking quality (CQ) and consumption, are controlled by physicochemical properties of starch in rice grain endosperm [1].

The ratio of amylase to amylopectin as well as the structure of amylopectin itself can vary greatly between different rice genotypes [2]. Generally, grains with higher amylose content present a harder non-sticky texture after cooking, being preferred in several countries. Such feature is usually evaluated during grain development in different cultivars [3]. However, the genetic events that lead to this type of grain are not well understood, and genotypes that deliver such grains are not easily obtained. That is the reason it is so important to understand the behavior of grain-quality-related genes, which enable more efficient and precise breeding applications. 

The 27 known *Oryza* species span over 15 million years of evolution which we can take advantage of, since it constitutes a rich source of genetic variation. Though a better understanding of the genomic differences between these species is essential for such purpose, the recent publication of the genomes of 13 rice species has opened the door to a series of new studies that make it possible to enrich the germplasm that can be used for breeding [4,5]. The possibility of using these wild species to improve grain quality should also be considered, but what would be the first genes to start such an analysis? 

The answer to this question is directly linked to the formation of amylose and amylopectin, which are two types of polysaccharides that form starch granules, and both have very complex biosynthesis processes. The ratio of these polysaccharides has a major influence on the appearance and structure of starch granules and consequently affects the quality of food production and industrial applications [1,6]. *Starch Synthesis-Related Genes (SSRGs)* are involved in this complex starch biosynthesis process, and can be divided into four classes: ADP-glucose pyrophosphorylase (AGPase), starch synthase (SS), starch branching enzyme (SBE) and starch de-branching enzyme (DBE). A very simplified explanation can be described, and this begins in cytosol for synthesis of amylopectin, in which AGPase synthesizes ADP-glucose from Glc1P and ATP through AGPSs (AGPS2a), which plays a catalytic function, while AGPLs (AGPL1, AGPL3, AGPL4) are mainly responsible for modulating the allosteric regulatory properties [7,8]. Already in plastid, the elongation of glucan chains occurs through 8 SS enzymes: SSI plays a role in short chains; SSII1/2/ALK in medium chains; SSIII1/2 in long chains; and SSIV1/2 in terminal chains for formation of 1,4-α-D-glycosyl [9,10]. Meanwhile, the other SS enzymes, GBSSII and Waxy, synthesize amylose (10). After the formation of 1,4-α-D-glycosyl, the enzymes from SBE class, SBE1 and SBE3 [9], work by branching the chains until the formation of 1,6-α-D-glycosyl. From there, the enzymes from DBE class—PUL [11], ISA [12] and DPE1 [13]—de-branch short external chains of glucans and also influence the activity of α-amylase and β-amylase [6].

In the first studies with SSRGs, it was identified that the Waxy gene (Granule-bound starch synthase I—from SS class) is directly involved with AC [13], while ALK (Starch synthase II-3—from SS class) is involved with TG [9]. The following studies contributed to the understanding that each SSRG is involved with one of the main starch-related quality characteristics (AC, GT, CQ) [9]. Despite this, it is also known that all SSRGs act in a coordinated way, forming a fine regulatory network in which the absence or change in the performance of a gene can ultimately lead to grain malformation [6,14], making it difficult to predict the nature of the starch resultant from biotechnological modifications. This makes it necessary to gain more in-depth knowledge about each of these genes through Targeted and Open-Ended Approaches [15]. Some of these have already been studied in some rice genotypes, achieving high-amylose content by: (1) transgenic knockdown of SBEI, SBEIIa and SBEIIb [16]; (2) CRISPR/Cas9-Mediated targeted mutagenesis of Starch Branching Enzymes [17]; (3) editing of rice isoamylase gene ISA1 [18]; and (4) downregulating of SSII2 caused lower AC in the endosperm [19]. Despite that, many studies regarding the variation of isozyme show that some variations in gene structure may not be so beneficial, mainly the non-synonymous substitutions that can affect the active domain of the isozyme. As presented through TILLING, in which mutations in the region of the exon 9/intron and exon 10 caused AGPL subunit (AGPase class), mutants severely shriveled with low weight and starch content [7]. 

Considering the importance of SSRGs in the control of CQ and the limited exploration of the information recently made available to the scientific community on *Oryza* genomes, an evolutionary analysis is needed to reveal the role of adaptive mechanisms before and after rice domestication. It will thus help to understand the complexity of the evolution of enzymes involved in the starch synthesis pathways, and further provide the basis for approaches that can generate new phenotypes through new strategies to modify starch synthesis. We therefore selected a set of SSRGs according to Zeng et al. [20], to explore their molecular evolution across the genus *Oryza* in which we observe differences in the structure of genes and proteins that can imply changes in the content of amylose and amylopectin. Appendix A presents all SSRGs identified in 11 *Oryza* species and *Leersia perrieri*, which is the nearest outgroup of the genus *Oryza*.

## 2. Results

### 2.1. AGPase Subunits

The phylogenetic tree of the ADP-glucose pyrophosphorylase (*AGPase* genes), which includes both large (AGPL1, AGPL3, AGPL4) and small (AGPS2a) subunits, identified 48 genes across the *Oryza* and the outgroup (*L. perrieri*), which revealed three different groups (Figure 1 and Figure 2). The first group is formed by the *AGPS2a* genes with both large exon and intron structures (Figure 2B). This group is believed to have the highest similarity to the ancestor of every *Oryza AGPase* gene. *O. meridionalis* (AA) also contains the same large exon structure in the second group formed by *AGPL4* and, likewise, in the third clade which is a mixture formed by *AGPL3* and *AGPL1*, respectively (Figure 2A).

Positive selection pressure was identified in the *AGPS2a* gene (Figure 2B). The conserved motifs of the four analyzed AGPase subunits form a signature pattern, revealing that motifs 9 and 10 are not detectable in the NTP transferase domain of AGPS2a; the same occurs for motifs 8 and 9 in AGPL4; 7, 8 and 9 in AGPL1; and 3, 6, 8 and 9 in AGPL3, which are not found in some *Oryza* species (Figure 2C,D).

Recombination analysis based on the alignment does not show any evidence of recombination in the AGP partition. On the other hand, positive selective pressure (dN/dS > 1) was detected in some sites of sequence alignments, suggesting diversifying selection (Appendix A).

In relation to gene position, all *Oryza* species have *AGPS2a* positioned in Chr. 8; of note, we observed that *OMERAGPS2A* and *ONIVAGPS2A* are located on two different chromosomes (i.e., Chr. 9 and Chr. 4, respectively (Figure 1)). Possible differential Mobile Element Insertion (MEI) events related to these loci was investigated, a region of 50 kb up- and downstream of these genes were aligned, showing high similarity between *OMERAGPS2A* (AA) to *AGPS2a* of other species, which means that this change in position probably did not occur through transposable element (TE) insertion (Appendix A).

### 2.2. Starch Synthesis (SS) Genes

A total of 92 protein coding starch synthase (*SS*) genes (SSI, SSII1, SSII2, ALK, SSIII1, SSIII2, GBSSII, Waxy, SSIV1 and SSIV2) were found across the 12 genomes data set, while its phylogenetic analysis allowed the identification of nine different clades based on sequence similarity. Clades I, II, III, IV, V, VI, VII, VIII and IX typically represent SSIV1, SSI, Waxy, SSIII1, SSIII2, GBSSII/ALK, SSII2, SSII1 and SSIV2, respectively (Figure 1 and Figure 3).

Some genes that have long exons near the 5′ or 3′ UTRs, as observed in few SS proteins of *L. perrieri*, *O. longistaminata*, *O. brachyantha* and *O. meridionalis*, seem to be ancestors of other species SSs (Figure 3A,B). The duplication of gene *OMERSSIV1_2D* (Clade I) is a probable result of a sub-functionalization since it does not contain motifs 4, 7 and 8, which represent the catalytic domain of starch synthase (Glyco_transf_5) and (Glyco_transf_1) (Figure 3C,D).

In addition to *OMERSSIV1_2D*, another recent duplication was identified in the *O. meridionalis SSIII1* gene, but in this case both the original and duplicated copies look functional, containing all the motifs that are part of its characteristic domain. However, the large size of *OMERSSIII1_1D* (7844 bp longer than the original copy) is something that deserves more investigation, especially when we take into account the highly conserved profile of these genes (Figure 3A–C). It is also important to notice that the same large domain occurs in duplicated copies of SSII2 and Waxy in the outgroup *L. perrieri*. 

Some *Oryza* species and *L. perrieri* show changes in chromosome position of the *SS* genes relative to *O. sativa* (Figure 1), such as *OMERSSII1* from Chr. 10 to 4 (Appendix A), *OMERSSII2* from Chr. 2 to 6, *ONIVSSII2* from Chr. 2 to 6 (Appendix A) and *OLONSSIV2* from Chr. 5 to 9 (Appendix A). An alignment analysis shows that, for *OMERSSII1* and *OMERSSII2*, the change did not occur through a differential TE insertion, since an analysis of 50kb upstream and downstream of each gene shows a lack of or just partial synteny (fragments from approximately 40 Kb) between the other *Oryza* loci. In case of partial synteny, a significant presence of TEs in this region was not identified using that with RiTE-DB. Interestingly, *OMERSSII2* contains an inverted region of 50 kb that denotes an unusual rearrangement by translocation and inversion of blocks up- and downstream of the gene (Appendix A).

Recombination events are found in both *ALK* and *Waxy* gene copies (Appendix A), while for *Waxy* stronger evidence of breakpoints can be identified (Appendix A). However, the same was not observed for the other *SS* copies, where 117 were found to be under positive selection (Appendix A) with no recombination events detected. 

### 2.3. De-Branching Enzymes (DBE)

The *DBE* (De-branching enzymes) genes are classified as *DPE1*, *PUL* and *ISA*. In total, 35 *DBE* genes were identified in *Oryza* and *L. perrieri* in the 12 genomes data set (Figure 1 and Figure 4). The phylogenetic analysis showed that the DBE proteins can be grouped in two clades, one that comprises DPE1 (Group I) and the other consisting of a mixed group composed of PUL and ISA proteins (Group II) (Figure 4A).

*DPE1*, despite forming a conserved clade, presents some variations in its two subgroups. First, *O. meridionalis* (AA) shows the longest gene structure, with more than eight exons, being the longest in its 5′ UTR, something that contrasts with the usual short structure of *DPE1* genes (Figure 4B). Furthermore, *OMERDPE1*, *OBARDPE1* and *OLONDPE1* lack motif 9, which is part of glycoside hydrolase family 77 domain (Glico_transf_77), a domain responsible for cleaving the starch granule into smaller glucan molecules (Figure 4C,D).

On the other hand, *ISA,* different from *PUL*, contains long and frequent introns in its gene structure; besides this, it also possesses every single motif that forms the formerly discussed protein signature (Figure 4B,C). We identified an event in *O. glaberrima* where *PUL* (Figure 1) is duplicated and translocated from Chr. 4 to Chr. 6 (Appendix A). Recombination analysis was performed, but no recombination could be inferred (Appendix A).

### 2.4. Starch Branching Enzymes (SBE)

In total, 24 SBE (Starch Branching Enzymes) genes were identified in *Oryza* and *L. perrieri* (Figure 1 and Figure 5). According to the position of *L. perrieri* in the phylogenetic tree, *SBEs* are defined as a mixed clade, which presents a very conserved gene structure and protein signature that comprises *SBE3* and *SBE1* (Figure 5A–C). Although the conserved motif analysis showed that motif 9 is not present in *OPUNSBE3* and *OMERSBE3*, they contain many more exons than the *SBE3s* of other *Oryza* species (Figure 5C). 

SBE proteins are characterized by a modular architecture composed of an N-terminal domain with a carbohydrate-binding module family 48 (CBM48), a central α-amylase domain, as well as a α-amylase C-terminal domain (Figure 5D).

Positive selection can be seen through the alignment of these genes, with 64 sites undergoing a diversification process (Appendix A), but no recombination events were detected.

## 3. Discussion

### 3.1. AGPase Subunits 

In *Oryza* and other plants, the AGPase protein subunit is characterized by a core region that is important for catalytic activity, called the nucleotidyl transferase domain (NTP transferase) that is important in providing the substrate for starch biosynthesis. The absence of specific motifs can affect the endosperm starch synthesis limiting the reaction converting Glucose 1-Phosphate (Glc-1-P) and Adenosine triphosphate (ATP) to ADP-glucose and inorganic pyrophosphate (PPi) in amyloplasts, directly reflecting the control of carbon flux into the starch accumulation pathway, consequently causing a shrunken endosperm in rice [1,6,21].

The evolution of the subunits of ADP proteins in *Oryza* is notably different from that reported in the literature for counterparts in other plant species [22]. This is probably due to different rates of selective pressure between species, which makes even more complex the study of the diversification of *AGPS2a* (Figure 2B).

In other plant species, such as dicots, the small subunits are under higher purification selection, thus remaining more conserved over time than the large subunits, which are primarily responsible for most of the diversification of *AGPase* genes [22,23]. The large subunits concentrate most of the positive selection, showing a great variability. However, in our results, when comparing the *AGPase* copies only inside *Oryza* genus, the opposite is observed. One explanation for this would be that in *AGPase* genes the large subunits concentrate most of the duplications [24]. However, Non-Homologous Recombination (NHR) cannot yet be ruled out, since both our data and previous reports indicate that NHR can be more frequent than MEI in *Oryza* species [25]. On the other hand, contrasting evolutionary patterns are expected between paralogues, and in the case of *AGPase* in *Oryza*, some duplications are already known and accompanied by a change in cell compartmentalization (from plastids to cytosol) and in their regulating properties [26].

In relation to gene position change, Non-Homologous Recombination (NHR) is likely to have occurred, placing this *OMERAGPS2A* large block (upstream + gene + downstream) in Chr. 9. The locus from *O. nivara* in Chr. 4 has only a small ortholog block that corresponds to the end of the upstream region and the start of the downstream region. Small up- and downstream fragments similar to specific LTR-TEs were found using the Rice Transposable Elements database (RiTE-db), but it is unlikely that these are responsible for a translocation event. As previously reported, the most frequent events responsible for changing copy number variations and gene position to other chromosomes are mediated by either transposable elements, through MEI or NHR, for both *Oryza* and *Arabidopsis* [23,27].

### 3.2. Starch Synthesis (SS) Genes

The phylogenetic analysis showed that, in most *Oryza* species, SS isoforms have undergone different degrees of gene duplication, something that is also observed in most plant species. *Oryza* clades I, IV, V, IX possess a different genetic origin from clades II, III, VI, VII and VIII and, since paralogous genes tend to slowly accumulate variations over time, it is easy to notice a large variation when we compare SS genes between these two clades [28,29,30,31]. The distinct spatial pattern of starch deposition within a caryopsis, which is also related to differences in the temporal expression pattern between early (SSIII1, SSII2, GBSSII) and late (ALK, SSIII2, Waxy) expressed genes [10], is probably the result of variations accumulated over time. Overall, the phylogenetic tree analysis reveals a highly conserved structure for both gene and amino acid sequences, suggesting a strong evolutionary relationship between species in each SS.

Taking into account that sequence variation in SSRGs have a great influence in rice amylose content, gelatinization temperature and amylopectin chain length [32], although important, it is hard to understand the roles of each SS isoform in each of the characters, due to the high sequence variation among these genes. Furthermore, it is even more complicated when we consider its diversity of genes in starch biosynthesis. The structural features of the genes and duplicated copies denote that wild *Oryza* species can be used as a rich source of variability that can improve starch quantity and quality, mainly through modifications of amylopectin synthesis chains [1].

Expressed specifically in the developing rice endosperm and leaves, SSIII 1 and 2 include three other repeated domains in addition to the starch synthase domain. An N-terminal Carbohydrate-Binding Module (CBM) domain is a contiguous amino acid sequence within a carbohydrate-active enzyme with carbohydrate-binding activity (Figure 3C,D). Although no lack of protein motifs was observed that could affect the catalytic domain in SSIII, in *O. sativa* this domain synthesizes long chains, and a deficiency in SSIII1, that is, the second major enzyme [33], can indirectly enhance both the SS-I and GBSS-I gene transcripts. On the other hand, a survey of amino acid motifs of SS isoforms reveals that certain motifs are absent in certain *Oryza* species, as it is possible to notice in OsINDSSIV1, OLONSSIV1, OBARSSII2 and OMERSSII2, which are part of the two C-terminal domains. This may affect the catalytic performance of the chain-elongation reaction of α-1-4-glucosidic linkage, which can further complicate the interplay between SS, SBE and DBE [34,35].

Waxy is believed to be the main enzyme that controls high amylose content in *Oryza* species and, with GBSSII, presents tissue-specific expression in a complementary manner between endosperm and non-endosperm tissues, causing different characteristics with respect to amylose content, and branch length distribution in amylopectin [36]. Thus, the differential action of these two enzymes affects the final amylose content in the endosperm. Despite this, the absence of GBSSII (Appendix A) does not influence the high content of amylose in the endosperm (about 35%) of *O. meridionalis* [37]. Despite the evolutionary advantage that the presence of the two enzymes (Waxy and GBSSII) confer for starch biosynthesis, Waxy enzymes without GBSSII seem to be enough for high amylose accumulation in *Oryza* endosperm, something that brings new perspectives for the improvement of this complex network [15,36,38].

On the other hand, the loss of SSIV2 in *O. glaberrima* during evolution does not eliminate the ability of chloroplasts in producing starch granules, since features in the N-terminal extension of SSIV enable the interaction with other proteins contributing to granule initiation. In *Arabidopsis*, when the SSIV glucosyl transferase domain is absent, a significant reduction of starch synthesis is observed [39,40].

The only *SS* genes that show evidence of having undergone recombination are *ALK* and *Waxy*. This agrees with previous reports, in which the diversification in *SS* genes was suggested to be driven by many duplication events instead of recombination events [41].

### 3.3. Debranching Enzymes (DBE)

DPE1 is a protein part of Group 1, which comprises enzymes that act in the initial phase of endosperm development [9], playing an important role in grain quality improvement programs [20]. A total absence of DPE1 was observed in *O. nivara* and *O. brachyantha*. Although there is not much clarity about the performance of DPE1 in *Oryza* chloroplasts, it is known that *Arabidopsis* plants lacking the plastidic DPE1 accumulate maltooligosaccharides (maltotriose-maltoheptaose), but not maltose, an important carbohydrate in starch formation [42].

Completely different from DPE1, regarding its phylogenetic position and structure, but also showing an important influence in the final portion of the starch synthesis pathway in *Oryza*, the enzymes PUL and ISA catalyze different reactions, but both have a conserved gene structure. Although they play unique roles in regulating the crystallization and degradation of starch, the enzymes have a close relationship in *Oryza* and share, as expected, the N-terminal O-Glycosyl hydrolase (CBM_48) and central domain alpha-amylase (Aamy), in which both degrade amylopectin. However, in some species like *O. sativa* v.g. *japonica* and *O. longistaminata*, there is still an absence of the C-terminal domain DUF_3372 domain (Figure 4D), which characterizes the Pullulanase, and usually cleaves the α-1,6-linkages of polyglucans in pullulan. This absence may affect the final endosperm amylose content. The main gene that controls amylose is *Waxy*, but as starch synthesis is a fine regulatory network, together with other enzymes like *PUL*, *AGPase*, *SSI*, *ALK*, and *SSIII2,* they control the final content of amylose (AC). However, in the absence of pullulan degradation, the final starch content may be lower, and consequently the AC is lower as well [15]. Exactly what is perceived in the *O. sativa* ssp *japonica* genotypes is that they have amylose content around 10–22% (low AC) while *O. sativa* ssp *indica* show 18–32% (high AC) [43,44].

Regarding the events of changing gene positions in chromosome of *DBE* genes, NHR constitutes a relatively frequent event in *Oryza* genomes, one might think that MEI could also be the responsible for such duplication and translocation, since these events frequently generate syntenic failures between homologue chromosomes when comparing different species [45]. Here we show (Appendix A) that it is not possible that MEI insertion could have occurred in these *PUL* genes. The same event could also have occurred in the other genes that have different chromosome positions (Figure 1). No recombination inference was found. However, 43 sites were observed to be under positive selection (Appendix A) in *DBE*, *ISA* and *PUL* gene copies phylogeny. Both Nougué et al. [41] and Qu et al [6] reported *DBE* homologue diversification through the detection of strong positive selection over these genes, once again denoting the complex evolutionary history of starch biosynthesis pathway.

### 3.4. Starch Branching Enzymes (SBE)

*Oryza* species present multiple SBE isoforms, more than shown here, but these are the major genes involved in the synthesis of amylopectin [20]. In modular architecture, both C and N termini play important roles in determining the substrate preference, catalytic capacity and chain length transfer [46]. The importance of SBE1 in synthesis of B1, B2, B3 chains of amylopectin has been reported in rice mutants [47,48], while others show that SBE3 has a role in the synthesis of 1–6 branching linkage [49]. In *Oryza*, these two enzymes are in the same clade. Some residues of binding sites for maltopentaose and glucose were not conserved between SBEI and SBEII isoforms; however, these residues were mainly found in SBEIII, which seems to be the reason for such a close proximity between SBE1 and SBE3 in *Oryza* [6].

## 4. Materials and Methods

### 4.1. DNA/RNA Sequencing and Gene Prediction

DNA sequences used in this study were obtained from the complete genome sequencing of *O. rufipogon* (Cultivar: W1943; Gramene accession: PRJEB4137), *O. nivara* (IRGC:100897; AWHD00000000), *O. glumaepatula* (GEN1233; ALNU00000000), *O. glaberrima* (IRGC:96717; ADWL00000000), *O. barthii* (IRGC:105608; ABRL00000000), *O. meridionalis* (OR44 (W2112); ALNW00000000), *O. punctata* (IRGC:105690; AVCL00000000), *O. brachyantha* (IRGC:101232; AGAT00000000), *O. longistaminata* (unnamed accession; PRJNA545798) and *Leersia perrieri* (A. Camus) Launert (IRGC:105164; ALNV00000000). Sequencing, assembly and annotation of these species are part of the IOMAP initiative in which Next Generation Sequencing (NGS) was used for obtaining both genomic and transcriptomic sequences.

### 4.2. Identification of SSR Genes in the Oryza Genus

The Starch Synthesis-Related (SSRGs) used in this study were chosen according to Zeng et al., [22]. Initially, the SSRGs of *O. sativa* ssp. *japonica* were obtained through the RAP-DB (The Rice Annotation Project Database) (https://rapdb.dna.affrc.go.jp/index.html (accessed on 29 April 2021)). The similarity of these SSRGs of *O. sativa* ssp. *japonica* to sequences of other *Oryza* species was evaluated through the BLAST tool [50], available in the ENSEMBL PLANTS database (http://plants.ensembl.org/index.html (accessed on 29 April 2021)). Only SSRGs that had the high score (values > 0) associated with high coverage (values > 0) and low e-value (value ≤ 0) were selected for analysis (Appendix A). Duplicated genes were identified with these same parameters but in this case in other locations in relation to the original gene. All genes and protein SSRG sequences are available in Appendix A.

### 4.3. Phylogenetic Analysis

The 19 resulting SSRGs were subjected to ClustalW [51] global alignment, generating an initial tree built through Neighbor-joining method [52] with 10,000 bootstrap replicates, with the aid of the Molecular Evolutionary Genetics Analysis 7—MEGA 7 [53] using the Gonnet matrix. The best replacement model was obtained through analysis in MEGA7. The appropriate model was selected for use in Bayesian analysis using Bayesian Evolutionary Sampling Trees—BEAST [54] with 1,000,000 bootstrap replicates. The resulting tree was plotted in FigTree (http://tree.bio.ed.ac.uk/software/figtree/ (accessed on 29 April 2021)). *Leersia perrieri* was used as the outgroup. The ratio between non-synonymous and synonymous substitutions (dNS/dS) for each SSRG was estimated using the Single Likelihood Ancestor Counting (SLAC) method, to infer the evolutionary force at work in each Open Reading Frame (ORF). The MEME test (Mixed Effects Model of Evolution) was applied to detect branches that are proportionally in higher positive selection pressure based in the Likelihood ratio test for episodic diversification (LRT). To evaluate the presence of recombination in each gene partitions used in this study, the Genetic Algorithm for Detection of Recombination (GARD) was applied; such recombinant sequences can cause misinterpretation in the phylogenetic relationships because recombination selection inference often leads to a significant increase in false positives. All analyses are available at Datamonkey [55] with *p*-value threshold of 0.1. The AUGUSTUS annotation software [56] was used to construct the structure of *SSRGs* and for GSDS 2.0 visualization [57]. The identification of the SSRG protein motifs was performed using Multiple Motif In Elicitation version 4.11.1 (MEME; http://meme-suite.org/tools/meme (accessed on 29 April 2021)) [58], considering the maximum number of motifs equal to 10. Protein domain analysis was performed in SMART database (http://smart.embl-heidelberg.de/ (accessed on 29 April 2021)) [59].

### 4.4. Translocation Event Analysis

To understand the origin of *ONIVAGPS2a*, *ONIVISA*, *ONIVPUL*, *ONIVSSII2*, *OGLAPUL_2D*, *OMERAGPS2a*, *OMERISA*, *OMERSSII1*, *OMERSSII2*, *OLONAGPL4* and *OLONSSIV2*, which could have occurred due to translocation events, we performed the alignment of regions corresponding to 50 Kb upstream and 50 Kb downstream from these genes in all the analyzed *Oryza* species, using Mauve [60]. We also used the RiTE database [61] to verify if this possible event occurred due to the translocation of transposable elements (TEs).

### 4.5. Chromosome Position Analysis

A circular MapChart-based plot was created in which the location of each gene can be seen. For the correct positioning of the gene of each species on the respective single/common chromosome, a simple percentage calculation was performed in order to establish a proportion relation when comparing the location of the genes in homoeologous chromosomes, according to the following equation:$$\frac{\mathrm{gene}\;\mathrm{location}\left(\mathrm{bps}\right)\times 100}{\mathrm{chr}\;\mathrm{size}\left(\mathrm{bps}\right)}=\mathrm{Relative}\;\mathrm{position}\;\mathrm{of}\;\mathrm{the}\;\mathrm{gene}\;$$

## 5. Conclusions 

In summary, we identified and characterized *SSRG* homologs in the wild relatives of Ρrice. Using phylogenetics and comparative genomics analyses we offer insights for the use of their gene variations in plant breeding. We confirmed the relative conservation of SSRGs between species within the AA-, BB- and FF-genomes, but structural analysis of these proteins suggest that deletions/mutations of amino acids in some active sites can result in structural variation that may negatively affect specific phases of starch biosynthesis. Direct modification of the endosperm, as usually observed in *O. sativa* ssp. *japonica*, which possesses lower AC, can likely be related to the absence of PUL C-terminal domain. The complete deletion of some genes appears not to affect the final composition of starch in the endosperm, as observed for *GBSSII* in *O. meridionalis*, *SSIV2* in *O. glaberrima,* and *DPE1* in *O. brachyantha* and *O. nivara.*

The analysis of structural features points to both absence of and duplicated copies of some motifs that can modify metabolic activity, denoting that the use of different *Oryza* species can be a rich source of variability for starch-targeted improvement in rice. These genes should now be further investigated by phenotyping different mutants and through the characterization of starch content of both wild *Oryza* genotypes and near isogenic lines (NILs) of *O. sativa* containing introgressions of these wild relatives. Such an analysis will help us to reveal the role of each variation of these genes, thereby contributing greatly to the simplification of the improvement processes that involve this complex path.

## Figures and Tables

**Figure 1 plants-10-01057-f001:**
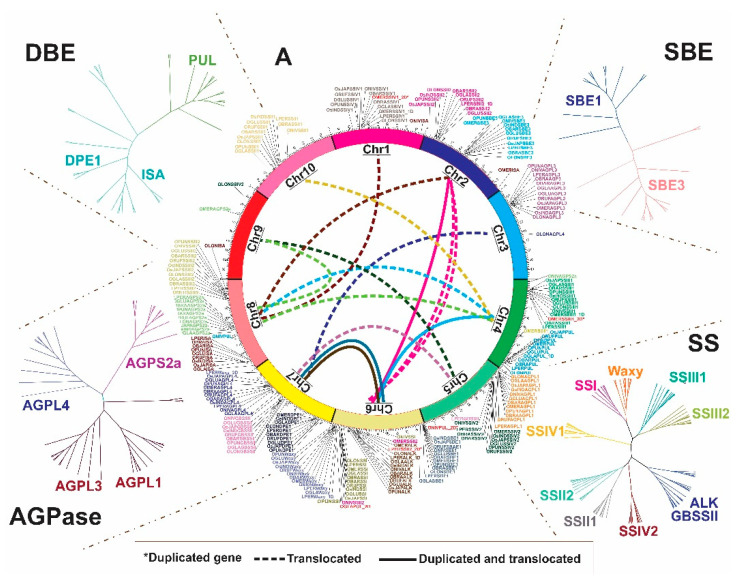
Gene localization and phylogeny of SSR proteins. Center. Circular map of SSRGs in a single representation from chromosomes 1 to 10 in *Oryza* species. Duplicated genes are marked with an asterisk. Around the map. Phylogenetic analysis of DBE (De-branching enzymes) (represented by DPE1 (Disproportionating enzyme), ISA (Isoamylase), PUL (Pullulanase); SBE (Starch branching enzymes) (represented by SBE1 and SBE3); SS (Starch synthase) (represented by SSI, SSII1, SSII2, ALK (Starch synthase II-3), SSIII1, SSIII2, SSIV1, SSIV2, Waxy (Granule-bound starch synthase I), GBSSII (Granule-bound starch synthase II); and AGPase proteins. Clade groups are indicated by different colors followed by the name.

**Figure 2 plants-10-01057-f002:**
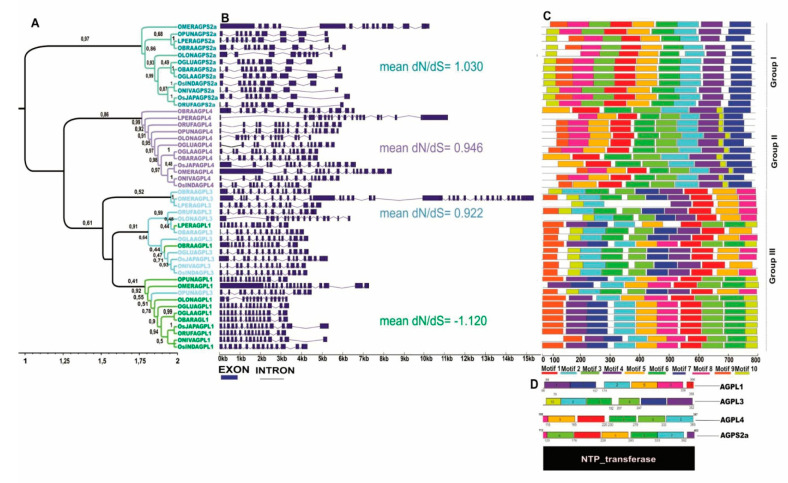
Phylogenetic relationship, genetic structure and analysis of conserved motifs in *AGPase* genes of species of the genus *Oryza*. (**A**) Phylogenetic protein tree. The branches of the AGPS2a, AGPL4, AGPL3 and AGPL1 proteins are marked in dark green, purple, light blue and light green, respectively. The bootstrap values are indicated in the phylogenetic tree. (**B**) Exon-intron structure of the *AGPase* genes in *Oryza*. (**C**) Protein motifs indicated in different colors. (**D**) Protein domain shared by AGPase proteins.

**Figure 3 plants-10-01057-f003:**
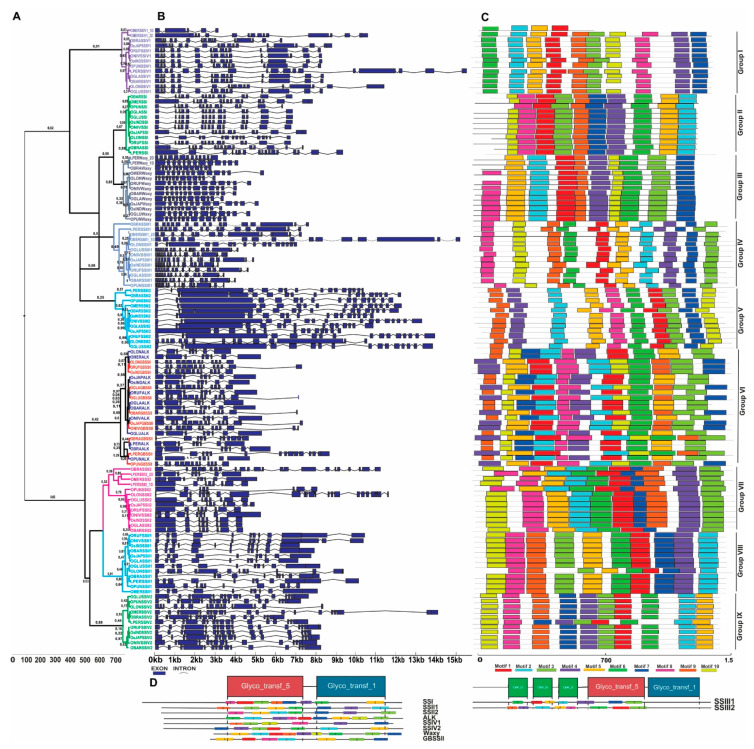
Phylogenetic relationship, genetic structure and conserved motifs/domain analysis in SS genes of *Oryza* species. (**A**) Phylogenetic protein tree and bootstrap values of SSI, SSII1, SSII2, ALK, SSIII1, SSIII2, GBSSII, Waxy, SSIV1 and SSIV2. (**B**) Exon-intron structure of SS genes in *Oryza*. (**C**) Arrangement of the 10 most frequent motifs found in the analyzed proteins. (**D**) Composition and distribution of domains and conserved motifs of SS proteins.

**Figure 4 plants-10-01057-f004:**
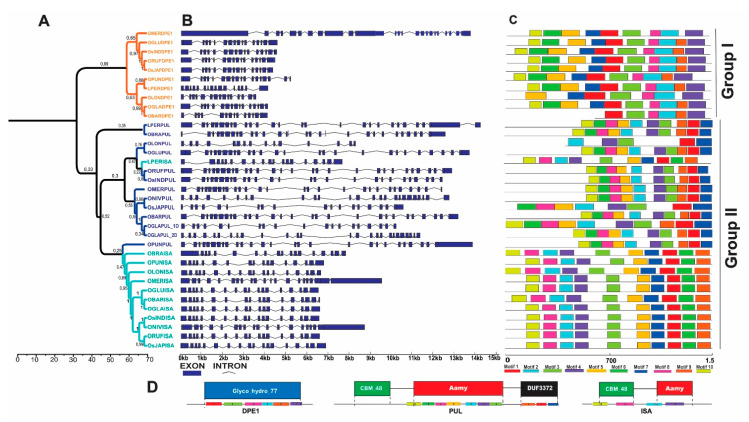
Phylogenetic relationship, genetic structure and conserved motifs/domain analysis in DBE genes of *Oryza* species. (**A**) Phylogenetic protein tree and bootstrap values of DPE1, PUL and ISA with branches marked in orange, blue and green, respectively. (**B**) Exon-intron structure of DBE genes in *Oryza*. (**C**) Arrangement of the 10 most frequent motifs found in the analyzed proteins. (**D**) Compositions and distributions of domain structures and conserved motifs of DBE proteins.

**Figure 5 plants-10-01057-f005:**
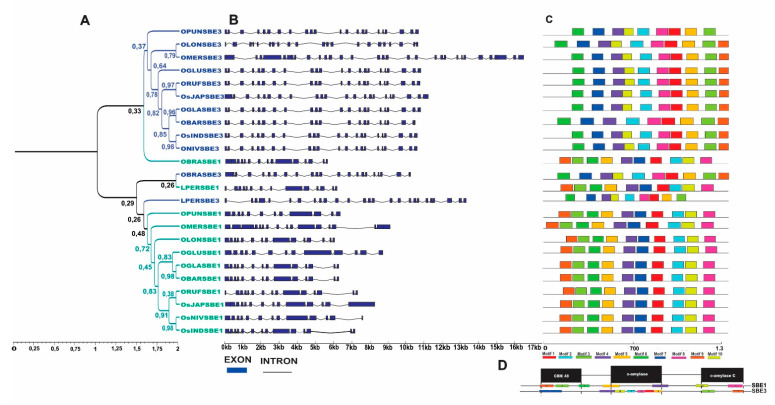
Phylogenetic relationship, genetic structure and conserved motifs/domain analysis in *SBE* genes of *Oryza* species. (**A**) Phylogenetic protein tree and bootstrap values of *SBE1* and *SBE3* with branches marked in green and blue, respectively. (**B**) Exon-intron structure of *SBE* genes in *Oryza*. (**C**) Arrangement of the 10 most frequent motifs found in the analyzed proteins. (**D**) Compositions and distributions of domain structures and conserved motifs of SBE proteins.

## Data Availability

Not applicable.

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
