# Peer review of "Starch Synthesis-Related Genes (SSRG) Evolution in the Genus Oryza"

_plants, 2021, doi:10.3390/plants10061057_

Round 1

Reviewer 1 Report

The paper describes results of analysis of starch synthesis related genes in the genus Oryza. In my opinion, its content is important and deserves publication. However, in my opinion, it contains numerous deficiencies, which require a major revision before it can be accepted for publication. Below I give few examples of the aforementioned deficiencies.

  1. Inconsistent formatting of Latin names, all of which, should be in italics.
  2. Page 2, line 69. When the Latin name is used at the beginning of a sentence it should be given in full.
  3. Page 3, line 73. Why the word “center” is repeated?
  4. Page 3, line 75. “Around, from the upper left, clockwise”. Wrong language.
  5. Page 4, line 89. “analysis based in the alignment do not show”. Should be: analysis based on the alignment does not show”.
  6. Page 5, line 111. “Another recent duplication”. I could not find any description of duplication(s) in the preceding paragraph on page 4.
  7. Page 5, line 125. Should be: synteny.
  8. Page 5, lines 129-130. I do not understand this sentence.
  9. Page 6, line 137. I do not understand this sentence.
  10. Page 7, lines 172-173. How exactly positive selection was found in the gene alignment?
  11. Page 8, lines 189-201. This part is very confusing. I am not sure what the authors wanted to say. Please, clarify. What “allies”? When an abbreviation is used it should be explained first. The abbreviation NHR is explained on line 202.
  12. Page 9, line 228. What species?
  13. Page 10, line 259. Wrong language.
  14. Page 10, 297. “positive selection under these genes”. I do not understand.
  15. Page 11, line 321. “of O. rufipogon”. Is it a new sentence? If so, it should start with an uppercase letter.
  16. Page 11, lines 352-357. Wrong language.
  17. Page 12, lines 365-371. Wrong language. “your” position?

Author Response

Reviewer 1:

  1. Inconsistent formatting of Latin names, all of which, should be in italics.

Latin names are now italicized.

  1. Page 2, line 69. When the Latin name is used at the beginning of a sentence it should be given in full.

 Corrected.

  1. Page 3, line 73. Why the word “center” is repeated?

Corrected.

  1. Page 3, line 75. “Around, from the upper left, clockwise”. Wrong language.

Corrected.

  1. Page 4, line 89. “analysis based in the alignment do not show”. Should be: analysis based on the alignment does not show”.

Corrected.

  1. Page 5, line 111. “Another recent duplication”. I could not find any description of duplication(s) in the preceding paragraph on page 4.

Here we referred to the duplication of OMERSSIV1_2D (previous paragraph). We tried to make it clearer now.

  1. Page 5, line 125. Should be: synteny.

Corrected.

  1. Page 5, lines 129-130. I do not understand this sentence.

This sentence says that recombination points were found in the ALK and Waxy genes, but that the evidence was clearer for the waxy genes. We tried to make it clearer now.

  1. Page 6, line 137. I do not understand this sentence.

This sentence refers to the composition and distribution of domains and conserved motifs found in SS proteins. We tried to make it clearer now.

  1. Page 7, lines 172-173. How exactly positive selection was found in the gene alignment?

This process can be seen in figure S11. Detection was carried out using Likelihood ratio test (LRT) that detect branches that are proportionally in higher positive selection pressure, described in the methods section. We try to make the text clearer.

  1. Page 8, lines 189-201. This part is very confusing. I am not sure what the authors wanted to say. Please, clarify. What “allies”? When an abbreviation is used it should be explained first. The abbreviation NHR is explained on line 202.

 We modified the paragraph to make the ideas clearer.

  1. Page 9, line 228. What species?

We refer to the wild Oryza species. We added this to the text.

  1. Page 10, line 259. Wrong language.

 Corrected.

  1. Page 10, 297. “positive selection under these genes”. I do not understand.

We have modified the sentence: “Both Nougué et al. [30] and Qu et al [8] reported DBE homologues diversification through the detection of strong positive selection over these genes, once again denoting the complex evolutionary history of starch biosynthesis pathway.”

  1. Page 11, line 321. “of O. rufipogon”. Is it a new sentence? If so, it should start with an uppercase letter.

No, that is not a new sentence, the period was placed incorrectly.

  1. Page 11, lines 352-357. Wrong language.

The MEME test (Mixed Effects Model of Evolution) was applied to detect branches that are proportionally in higher positive selection pressure. To evaluate the presence of recombination in each gene partitions used in this study, the Genetic Algorithm for Detection of Recombination (GARD) was applied, such recombinant sequences can cause misinterpretation in the phylogenetic relationships because recombination selection inference often leads to a significant increase in false positives.

  1. Page 12, lines 365-371. Wrong language. “your” position?

Corrected.

Reviewer 2 Report

The study is novel, but the flow of this article and how the authors presented their results must be improved. In the Introduction, the authors might need to introduce a bit of what is SSRGs and what genes included in the group, to give some idea to the reader before you go through the Results. In the Results, some scientific names are not italicized and moderate english changes required. Give reference for the equation in page. 12. 

Author Response

Reviewer 2:

  1. The study is novel, but the flow of this article and how the authors presented their results must be improved. In the Introduction, the authors might need to introduce a bit of what is SSRGs and what genes included in the group, to give some idea to the reader before you go through the Results.

We made several changes throughout the text as a way to facilitate the flow of reading of the manuscript.

  1. In the Results, some scientific names are not italicized and moderate english changes required.

Corrected.

  1. Give reference for the equation in page. 12.

This equation is a simple rule of three for maintaining proportions when using different chromosomes. There is no specific reference.

Reviewer 3 Report

In this paper, the authors have performed bioinformatics analysis of rice protein expression using publicly accessible NGS sequence data of different varieties.

The paper is easy to follow; I suggest only a few edits below.  

Suggested edits:

  1. The introduction is thin and the authors should augment it with additional information about the protein isozyme variation in rice including for amylase and the other proteins analyzed (AGPase, etc.) and how they affect the rice properties.
  2. Protein abbreviations should be defined at first use (e.g., Figure 1: DPE, PUL, AGPase, ALK, SBE, SS, ISA, TE). Some are defined later in the paper but I found myself searching for them to interpret Figure 1. In 2.1 Results, AGPase should be defined as ADP-glucose pyrophosphosphate in the first paragraph rather than the second one.
  3. All genus species names should be italicized (page 4 lines 106-107 etc.).
  4. Page 5, line 125 sinteny should be synteny as in line 123.

Author Response

Reviewer 3:

  1. The introduction is thin and the authors should augment it with additional information about the protein isozyme variation in rice including for amylase and the other proteins analyzed (AGPase, etc.) and how they affect the rice properties.

We added new information to the introduction to clarify the roles assigned to the genes under analysis.

  1. Protein abbreviations should be defined at first use (e.g., Figure 1: DPE, PUL, AGPase, ALK, SBE, SS, ISA, TE). Some are defined later in the paper but I found myself searching for them to interpret Figure 1. In 2.1 Results, AGPase should be defined as ADP-glucose pyrophosphosphate in the first paragraph rather than the second one.

Corrected.

  1. All genus species names should be italicized (page 4 lines 106-107 etc.).

Corrected.

  1. Page 5, line 125 sinteny should be synteny as in line 123.

Corrected.

Round 2

Reviewer 1 Report

I do not have new comments to this manuscript.

Author Response

 Thank you for your reviewing